# Scan Pattern Characterization of Velodyne VLP-16 Lidar Sensor for UAS Laser Scanning

**DOI:** 10.3390/s20247351

**Published:** 2020-12-21

**Authors:** H. Andrew Lassiter, Travis Whitley, Benjamin Wilkinson, Amr Abd-Elrahman

**Affiliations:** 1Geomatics Program, School of Forest Resources and Conservation, University of Florida, Gainesville, FL 32611, USA; benew@ufl.edu (B.W.); aamr@ufl.edu (A.A.-E.); 2Geospatial Modeling and Applications Lab, School of Forest Resources and Conservation, University of Florida, Gainesville, FL 32611, USA; 3Altavian Inc., Gainesville, FL 32601, USA; twhitley@altavian.com

**Keywords:** UAS, drones, lidar, flight planning, Velodyne

## Abstract

Many lightweight lidar sensors employed for UAS lidar mapping feature a fan-style laser emitter-detector configuration which results in a non-uniform pattern of laser pulse returns. As the role of UAS lidar mapping grows in both research and industry, it is imperative to understand the behavior of the fan-style lidar sensor to ensure proper mission planning. This study introduces sensor modeling software for scanning simulation and analytical equations developed in-house to characterize the non-uniform return density (i.e., scan pattern) of the fan-style sensor, with special focus given to a popular fan-style sensor, the Velodyne VLP-16 laser scanner. The results indicate that, despite the high pulse frequency of modern scanners, areas of poor laser pulse coverage are often present along the scanning path under typical mission parameters. These areas of poor coverage appear in a variety of shapes and sizes which do not necessarily correspond to the forward speed of the scanner or the height of the scanner above the ground, highlighting the importance of scan simulation for proper mission planning when using a fan-style sensor.

## 1. Introduction

Airborne laser scanning (or lidar) from small unoccupied aircraft systems (UAS) has enjoyed wide adoption in the fields of topographic mapping [1], agriculture [2], forestry [3], and urban mapping [4], to name but a few. However, this rapid adoption has outpaced careful study of the sensor’s accuracy and repeatability, leading to few publications on the topic [5,6]. Alongside sensor accuracy and repeatability lies another chief concern for the acquisition of UAS lidar mapping data: the resulting pattern of laser pulse returns, i.e., the scan pattern, of the multi-beam, “fan-style” lidar sensor often used in UAS lidar mapping (Figure 1). A thorough study of the scan pattern of the multi-beam lidar sensor (and the only study of its kind, to the authors’ knowledge) is presented in Morales et al. [7]; however, the problem is approached with the lidar sensor in a static, terrestrial pose. The scan pattern of the multi-beam lidar sensor, particularly from the dynamic, aerial pose (Figure 1), is the principal topic of the following study.

Mission planning for UAS lidar mapping is a new problem that is scarcely addressed in existing literature. The most basic considerations for mapping a target area with a given lidar sensor—e.g., flying height, swath width, and percent sidelap of adjacent swaths—can be deduced from trigonometry and arithmetic. However, the low altitude pose, coupled with the lightweight and relatively low-cost lidar sensors most commonly used for UAS lidar mapping, present new questions that must be addressed to ensure the quality of the lidar data collected.

Perhaps the most crucial aspect of a lidar mapping mission is point density. Methods for calculating point density are sensor-dependent—or, more specifically, are dependent upon the pulse rate of the sensor and the geometry of the sensor’s emitted laser pulses. A small number of solutions exist for calculating the expected point density of a UAS lidar mapping mission that account for specific model of lidar sensor as well as flight parameters such as flying height, forward speed, and percent overlap [8,9,10]. Another key aspect of a UAS lidar mapping mission is the relatively high variation in ranges to target and incidence angles of the laser pulses with objects in the scene. Consider a conventional airborne lidar mapping mission conducted from a piloted aircraft flown many hundreds of meters above ground level. With a typical swath width of 15 to 20° and a flying height on the order of hundreds of meters, the resulting variation of ranges and incidence angles is much lower than those exhibited in a UAS lidar mapping mission, with flying heights on the order of tens of meters and swath widths of 90° or more. Range and incidence angle are key components of the lidar error budget [11,12], and careful study must be made of the expected pattern of laser returns—i.e., the scan pattern—of any lidar sensor used for high-accuracy mapping.

The multi-beam “fan-style” lidar sensor presents the unique problem examined in this study. Take for example the Velodyne VLP-16 lidar sensor, currently one of the popular sensors in the UAS lidar mapping field. The VLP-16 scanner head is comprised of sixteen lasers in a “fan” of diverging channels, resulting in a 30° vertical field of view. This “fan” of lasers rotates about the scanner’s vertical axis for a 360° horizontal field of view [13]. This configuration of lasers is functionally similar to the VLP-16′s predecessors, HDL-64, which was designed for self-driving automobiles [14]. This fan-style configuration is present in other sensors used in UAS lidar mapping, for example the Velodyne HDL-32 [5], Quanergy M8 [15], and the Ouster OS-1-16 [16].

To ensure maximum coverage along a flight line when using a fan-style sensor in the aerial pose, the most sensible way to orient the scanner is on its side, such that the scanner’s vertical axis is parallel with the direction of travel (Figure 1). The resulting scan pattern is a series of hyperbolas (Figure 2). However, the resulting scan pattern does not produce a homogeneous distribution of laser returns (or points), and in some cases, can produce relatively large gaps in coverage (Figure 3). These gaps in coverage are not a function of point density, but rather point clustering, where many laser pulses are striking the target within very close proximity to each other. This stands in contrast to point dispersion, e.g., a series of laser pulses which produce a more homogeneous pattern across the target.

The VLP-16 emits lasers from sixteen channels oriented between −15° and +15° in 2° intervals from the scanner’s horizontal plane, or xy-plane, resulting in a 30° vertical field of view. The vertical angle ω of each channel is fixed and is defined as the counterclockwise angle of the channel with respect to the scanner xy-plane. The channels fully rotate about the scanner’s vertical axis, or z-axis, for a 360° horizontal field of view. The lasers are fired one at a time according to a precise timing sequence in which one laser is fired every 2.304 μs; after all sixteen lasers have fired, there is a recharge period of 18.43 μs [17]. Therefore, each laser firing has a unique time, and because the scanner head is in constant rotation, each laser firing has a unique azimuth α, where azimuth is defined as the clockwise angle of rotation about the + z-axis, with the + y-axis set to zero [17]. Assuming the configuration described above, over flat terrain, the resulting scan lines from the VLP-16 are affine hyperbolas; as the scanner travels forward parallel to the terrain, these overlapping sets of skewed hyperbolas result in areas of varying point dispersion (Figure 3). Laser pulses are emitted at a constant angular interval (or, put more precisely, emitted at a constant time interval while rotating at a constant angular velocity) along the scanner’s 360° field of view. Under the presented configuration with the VLP-16 turned on its side, point density along a nominally flat surface is a function of linear distance across the scanning profile. The point density, along with the point clustering, are the primary considerations for planning a UAS data collection mission with the VLP-16.

Maximizing not only the accuracy but also the optimal dispersion of the laser returns is crucial for detection of small features, especially for horizontal accuracy assessment of a UAS lidar mapping system. Direct comparison cannot be made between a lidar point cloud and point features in the real world; geometric modeling of surfaces must be utilized to compare lidar data and the scanned area [18,19,20] For horizontal accuracy assessment, virtual points can be constructed via the intersection of surfaces. For practical reasons, especially when not in the presence of common surfaces used for accuracy assessment such as roofs, these surfaces are of finite size. To properly model the surfaces, one must ensure that (1) enough pulses reach the surface to accurately model it [21] and (2) those pulses are within reasonable thresholds of range and incidence angle. Therefore, the scan pattern of a UAS sensor must be characterized before data collection to ensure the resulting data will meet the standards necessary to perform the accuracy assessment at hand.

This study presents the development and use of VLP-16 simulation software for to examine the scan pattern of the VLP-16 both qualitatively—i.e., manually examining simulated point clouds—and quantitatively, via spatial statistics. Characterization equations for the VLP-16′s point density, optimal separation of flight lines, and possible gaps in coverage are also presented. These parameters can be modified, which facilitate using the software and point cloud simulation for other fan-style scanning systems.

## 2. Materials and Methods

The primary objective of this study was to characterize and analyze the scan pattern of the VLP-16 as a typical fan-style lidar sensor. To this end, the authors derived characterization equations to predict point density and expected coverage gaps. The authors also created simulation software which models the sensor in the typical UAS pose (Figure 1) over a target plane, a proxy for flat terrain. Both the characterization equations and simulation software are generalized for those sensor and flight parameters that have the greatest effect on the resulting scan pattern, i.e., flying height, forward speed, rotation rate of the scanner head, and orientation of the scanner with respect to the target plane.

Field data for verification of the equations’ and simulation software’s output were collected on 15 June 2018. The UAS used is a DJI S1000 octocopter with a custom-built lidar mapping payload comprised of the Velodyne VLP-16 lidar sensor (Velodyne Lidar, San Jose, CA, USA) and the original equipment manufacturer (OEM) component equivalent to the NovAtel SPAN-IGM-S1 integrated Global Navigation Satellite System and Inertial Navigation System (GNSS/INS) paired with a NovAtel GPS-702-GG GNSS antenna and a secondary Garmin GPS18x GPS antenna (used for timestamping the lidar pulses) (NovAtel Inc., Calgary, Alberta, Canada). The position data collected by the INS underwent a post-processed kinematic (PPK) solution using a nearby static GNSS reference station to refine its accuracy. All GNSS/INS observations—the raw GNSS observations, the PPK-adjusted GNSS observations, and the raw INS attitude observations—were then processed in a forward/reverse, multi-pass, tightly coupled solution in Waypoint Inertial Explorer software to produce a navigation solution which adjusts and minimizes the errors of the position and attitude observations of the mapping payload onboard the UAS. A custom Python algorithm, using the tightly coupled navigation solution, then (1) transformed the raw Velodyne lidar data form spherical to Cartesian coordinates in the scanner frame as described in [22], then (2) performed direct georeferencing of the transformed lidar data using the method described in El-Sheimy [23]. Both the mapping payload and the custom algorithms used to generate the lidar point clouds in this study can be considered typical for this category of lidar data acquisition systems.

In addition to the comparison of the field data with the simulated data and characterization equations, this study also explored a spatial analysis of the simulated scan pattern data to further characterize its non-uniformity. Further detail of these methods is given below.

### 2.1. Analytical Characterization of VLP-16 Scan Pattern

The point density of the laser return pattern of the VLP-16, as a function of lateral distance from the flight line x, can be closely approximated by the point density function
(1)$$p\left(x\right)=\frac{l_{f}h\cos\alpha }{2\pi v_{z}\left(h^{2}\cos^{2}\alpha +x^{2}\right)},$$
where h is the height above ground, $l_{f}$ is the pulse frequency of the scanner (approximately 300,000 pulses per second), $v_{z}$ is the forward velocity of the airframe, and α is the angular difference between the scanner’s z-axis and the direction of travel (herein referred to as the scanner’s “yaw”). This equation is simplified under the assumption that the laser pulses are emitted at a uniform rate, although in practice, this is not the case. Each of the sixteen channels of the VLP-16 emits a pulse once per 2.304 μs ; after all sixteen lasers have fired, a recharge period of 18.43 μs follows. This assumption, however, has a negligible effect on the results, as demonstrated by comparing the equation’s results to the scan pattern simulation presented in the Results section.

The bands of gaps present in the scan pattern (Figure 3) occur at certain lateral distances from the flight line $x_{i}$, and can possibly occur at the distances found using the equation
(2)$$x_{i}=h\tan\left(\cos^{-1}\left(\frac{hr\tan\Delta \omega }{iv_{z}}\right)\right)\cos\alpha ,$$
where r is the rotation rate of the scanner head (typically 5–20 Hz), Δω is the angular separation between adjacent channels (2° for the VLP-16); and i is an integer, indicating the $i^{th}$ gap outward from the flight line at x=0. Note that the term for which the arccosine is taken in Equation (2) can be ≥1, which yields a complex value of $y_{i}$.

To assure a minimum density of laser returns along the profile of the laser return pattern of a mission with parallel flight lines, the maximum flight line separation w can be expressed as
(3)$$w=2\sqrt{\frac{l_{f}h\cos\alpha }{\pi p_{d}v_{z}}-h^{2}\cos^{2}\alpha },$$
where the minimum desired return density is $p_{d}$, expressed as points/m^2^. This equation is a further derivation of the point density function. Inspection of the plot of p(x) (Equation (1)) should be used to inform a sensible choice for a value of $p_{d}$.

The derivation of Equations (1)–(3) are given in the Appendix A.

### 2.2. Simulation of the VLP-16 Scan Pattern

To test the VLP-16 scan pattern in a controlled environment, software was created in the MATLAB programming language to simulate the laser return pattern of the scanner in the aerial pose. The scanner is modeled as a point $s_{t}$ at a user-specified height above a horizontal target plane passing through the origin, traveling at a constant speed (also specified by the user) along a vector parallel to the target plane. The modeled emitted laser pulses are modeled as lines passing through the scanner point toward the target plane. The “vertical” angles ω of the lasers (referred to as “channels” by the manufacturer) are fixed, as described above. The “horizontal” angles of the lasers, or azimuths α, are determined by the orientation of the scanner head at each epoch as it rotates about the scanner’s vertical axis. The direction of each modeled pulse in the scanner-oriented coordinate system (SOCS), and the subsequent rotation of SOCS into the mapping frame R (as shown in Figure 1), is found by
(4)$$r=R\left(r_{SOCS}\right)=\left[\begin{array}{ccc} 1 & 0 & 0\\ 0 & 0 & -1\\ 0 & 1 & 0 \end{array}\right]\left[\begin{array}{c} \cos\omega \sin\alpha \\ \cos\omega \cos\alpha \\ \sin v \end{array}\right].$$

For each laser pulse emitted at epoch t, the simulation passes the line parameters ${\vec{s}}_{t}$ (the position of which is calculated according to the forward speed and time t) and direction of simulated pulse r to a solver function, which finds the intersection of the line with the target plane, using the common algebraic method. The solutions of these line-plane intersections—a set of points, i.e., a simulated point cloud—are recorded and saved in a text file. This is a simplified sensor model that does not account for the interior geometry of the sensor [5], the manufacturer reported range accuracy of the sensor [22], or the incidence-angle-dependent range bias observed in [11]. For the purposes of this study, however, the simplified sensor model is sufficient to study coverage gaps whose dimensions are two to three orders of magnitude greater than the previously mentioned error sources.

The software takes as input key mission parameters with regards to the orientation and operation of the scanner: height above ground, forward speed, scanner head rotation rate, and yaw (i.e., the difference in the scanner’s +z-axis and the direction of travel). (Yaw is accounted for by generalizing the rotation matrix R shown in Equation (4)). The output is a simulated point cloud that is a sample of the laser pulse return that could be expected from a mission flown over flat ground under these mission parameters. The point cloud output is a comma-delimited text file, where each row represents one return, or point. For each point, the following attributes are attached: the mapping frame coordinates of the solved line-plane intersection, (X,Y,Z)*;* the azimuth of the scanner head when the pulse was fired, α ; the vertical angle, or channel, from which the laser was fired, ω ; the time at which the laser was fired, t ; the range of the laser return (i.e., the distance between the scanner point and the solved line-plane intersection), ρ ; and the direction in the mapping frame that the laser was fired, $r={\mathrm{dir}}_{X},{\mathrm{dir}}_{Y},{\mathrm{dir}}_{Z}.$

The scan pattern analysis and gap equation seek to detect and predict the location of these bands as a function of the mission parameters of flying height, flying speed, and the rotation rate of the scanner head (which can be adjusted anywhere from 5 Hz to 20 Hz). Predicting where these bands of high clustering occur as a function of mission parameters will lead to recommendations for optimal values for the parameters mentioned above, as well as side lap (i.e., the overlap of adjacent swaths).

### 2.3. Spatial Analysis of Simulated Data

The scan pattern, for all its peculiarities, is in fact a repeating pattern. This allows for the extraction of a narrow across-track profile or “representative profile” (Figure 4) which can be used for statistical analysis. The apparent clustering and dispersion of the points along the profile is analyzed by binning the representative profile of the simulated point cloud along the across-track axis and calculating the nearest neighbor index (Equations (4)–(6)) for each bin. The nearest neighbor index is a measure applied to point patterns that indicates whether the pattern is clustered, random, or dispersed. First, for each simulated return (point) in the bin $s_{i}$, its nearest neighbor and the distance between the two ($d_{\min}$) is found. This is accomplished by calculating a distance matrix (of dimensions n×n) for n points in the bin, and for each column in the matrix, the lowest off-diagonal value is retained. The mean of those n nearest neighbor distances is the observed nearest neighbor distance, or ${\bar{d}}_{\mathrm{obs}}$:
(5)$${\bar{d}}_{obs}=\frac{\sum_{i=1}^{n}d_{min}\left(s_{i}\right)}{n}.$$

Next, the expected nearest neighbor distance ${\bar{d}}_{exp}$ is calculated, which is a function of the area A of the bin:(6)$${\bar{d}}_{\exp}=\frac{0.5}{\sqrt{n/A}}.$$

Finally, the z-score (i.e., nearest neighbor index) for each bin is found:(7)$$z=\frac{d_{\mathrm{obs}}-d_{\exp}}{SE},$$
where $SE=0.26136/\sqrt{n^{2}/A}$.

The z-score provides a normalized measure of the point dispersion or clustering in the bin. For example, a z-score of ≤−1.96 (with its corresponding *p*-value of ≤0.05), indicates with 95% or greater likelihood that the point pattern is clustered, while a z-score of ≥+1.96 (p≥0.95) indicates a 95% or greater likelihood that the pattern is dispersed, i.e., approaching even spacing amongst points (Clark and Evans, 1954; O’Sullivan and Unwin, 2010). It should be mentioned here that this presented interpretation of the z-score is reflective of the source material; however, it is not appropriate for analyzing data of this nature (see the Discussion section).

The analysis of the scan pattern extends beyond its geometric pattern. The point cloud simulation software records other attributes of each simulated return, such as the position of the scanner, range from scanner, time, and direction of the simulated laser pulse. These attributes can provide a simulation of not only of point density and dispersion, but also the expected absolute accuracy of the points. The information that can be gleaned from these attributes—e.g., range and incidence angle—are key components of the lidar error budget, as mentioned in the Introduction. Error propagation within the scanner’s frame (i.e., error related to the scanner itself, relative to the scanner’s coordinate system, not accounting for error from the navigation sensor data) is essential to understanding the reliability of the resulting point cloud. For this study, the simulated returns within each bin are analyzed according to attributes unique to each return. Within each bin, histograms can be generated to show the distribution of the number of returns as a function of average range and average incidence angle for each bin. These two measures are directly proportional over the ideal target plane used in this study’s simulation; as the incidence angle of a laser pulse “increases” (i.e., departs from being normal to the target), so too increases the range that pulse must travel to reach the target. An analog to the accuracy of the points within a given scanned area can be visualized by finding the distribution of ranges and scan angles across the scan profile.

## 3. Results and Discussion

### 3.1. Single Swath

As expected and shown in the point density function equation, the point density of the scan pattern is inversely proportional both to the flying height and forward velocity of the scanner. These relationships are shown in Figure 5 and Figure 6, respectively. The simulations are in agreeance with the point density equation and, in most cases, the gap equation. There are, however, three limitations of the gap equation (Equation (2)) made evident through simulation:Any gaps present at or near nadir to the flight line (i.e., gaps at some low value of $x_{i}$) are not always reported by the gap equation.The severity of the gaps at some lateral distance $x_{i}$ is not reflected in the results of the gap equation. The equation only reports the possibility of gaps occurring at some lateral distance.At higher forward velocities, some significant bands of gaps may not be reported by the gap equation.


An example of these limitations is shown in Figure 7 (near nadir gaps sometimes not predicted) and 8 (high velocity gaps sometimes not predicted). In conjunction with the gap equation results, inspecting the point density histograms for each bin’s z-score (Equation (6)) reveals flaws in using the nearest neighbor index as a proxy for detecting gaps (see Discussion). As observed through simulation (e.g., Figure 7 and Figure 8), in the areas of the greatest degree of point clustering, the gaps in the scan pattern tend to present as either near-rhombuses, narrow and elongated perpendicular to the direction of flight. As the lateral distance from the flight line $x_{i}$ increases, the gaps will sometimes present as somewhat linear clusters of points, which have a less deleterious effect on the coverage of the area. Gaps near nadir to the flight line are elongated to the point of presenting similarly to a widely spaced linear scan pattern, especially as the flying height increases (and the eccentricity of the hyperbolas decrease).

### 3.2. Rotation Rate and Gaps

The results of the gap equation (Equation (2)) can be initially misleading. As the rotation rate increases, the number of predicted possible locations of gaps increases proportionally. In fact, as the rotation rate doubles, so too does the number of predicted gaps. The predicted gap locations are somewhat linear, occurring every x meters along the profile at some rotation rate r ; at rotation rate 2r, the distance between gaps becomes roughly x/2. But the severity of those gaps is not reflected in the equation. In fact, as the rotation rate increases, the severity of the gaps decreases notably, as shown in Figure 9. This study does not comment on the potential benefits of lowering the rotation rate of the scanner head, but the results do show that a lower rotation rate, especially when coupled with a higher forward velocity, can have detrimental effects on the quality of the resulting coverage.

### 3.3. Effect of Yaw Angle

Adding yaw, i.e., crabbing the scanner with respect to the direction of flight, leads to the coverage gaps becoming narrower along track and elongated across track, with respect to the gaps that would occur under similar conditions without crabbing. This is shown in Figure 10 and Figure 11. The primary deleterious effect of the coverage gaps—increasing the odds of missing small areas of interest, e.g., linear features such as sidewalks or power lines, or small areas of interest such as targets [18]—is negated with the addition of only a small amount of yaw. In other words, the yaw of the scanner leads to a more homogeneous pattern of returns. This has a minimal effect on the width of a single swath, as its width is a function of the cosine of the yaw angle. For example, even a yaw angle of 30° would only yield a $1-\cos30^{{}^{\circ}}=13\%$ reduction in the width of that swath. This finding aligns with the results of [7], which achieved an analogous result by developing an automatic tilting platform for the VLP-16 in a static, terrestrial pose.

### 3.4. Overlapping Swaths

With the exception of one-way mission plans along some corridor, such as a utility easement or transportation corridor, most lidar data collection missions using UAS will likely feature parallel, overlapping swaths of data. These overlapping swaths not only provide common targets in each swath which can be used for strip adjustment and accuracy assessment, but also can be used to “fill in” the gaps present in the VLP-16 scan pattern. Using the optimal flight line separation equation, it is possible to plan a mission in which a desired minimum point density is achieved across the mission area most efficiently. Figure 12 shows the resulting point density and point dispersion of a mission flown at 45 m flying height, 9 m/s forward velocity, with parallel flight lines spaced at 50, 68, and 88 m. These flight line spacings are the result of the flight line separation equation for values of $p_{d}=$ 180, 150, and 120 points/m^2^ respectively.

### 3.5. Spatial Analysis

The question remains of which (if any) z-score threshold is indicative of point clustering resulting in undesirable gaps in coverage. The z-score interpretation presented above is based on the distribution of expected nearest neighbor distances in a theoretically random distribution within each bin; the nearest neighbor index is used to estimate the probability that a distribution of points in a given area is clustered, random, or dispersed. The points in the VLP-16′s scan pattern are not random events, which undermines the probabilistic interpretation of the z-score. In fact, the z-scores appear to be fully relative, only useful for comparison within a given flight configuration. Inspection of Figure 9 shows that both the 20 Hz and 5 Hz flights have comparable peak minimum z-scores of −18 and −21, respectively. This belies the severity of the gap problem in the 5 Hz flight while overstating the gap issue in the 20 Hz flight, which is all but negligible.

Final considerations for mission planning include optimizing the expected incidence angles and ranges to the objects in the scene. Figure 13 shows the average absolute values of scan angles of each return within 1-m bins across the profile. Wider spacing of flight lines may lead to fewer flight lines needed to cover an area, but as the scan angle (and thus range) of the returns increases, so too will their error. Another thought is to simply exclude returns beyond a certain range during postprocessing of the point cloud for the sake of a higher accuracy across the resultant point cloud.

Figure 14 shows an example scan profile where the maximum range has been limited to 60 m, down from the manufacturer’s reported maximum range of 100 m. The profile width decreases drastically, but depending on the desired accuracy of the data, this may be advantageous. Note that the optimal flight line separation equation cannot be used if the maximum range is limited in this manner.

### 3.6. Recommendations for Mission Planning

The study reveals trends in which mission parameters may lead to coverage gaps. The gaps are more likely to occur as overlap decreases (or overlap is not present), forward velocity increases, and/or the rotation rate of the scanner head decreases. The first two relations make sense: overlap obviously helps fill in any potential gaps; as forward velocity decreases, point density increases. The change in the scan pattern with respect to rotation rate of the scanner head, is not so straightforward.

The gaps were shown to have more potential to provide degrading quality of coverage as the rotation rate of the scanner head decreases. Velodyne allows the user to adjust the rotation rate between 5–20 Hz. A lower rotation rate may lead to increased accuracy of the azimuth reading in the scanner frame and/or decreased power consumption, though this was not examined in this study. Concerns of gaps in coverage, especially with respect to flying height, forward velocity, and the size of objects of interest in the scene, should be considered alongside the potential benefits of decreasing the rotation rate of the scanner head. but random, back-and-forth yaw/crab of the airframe helps to fill in gaps.

The study also suggests that the maximum range of returns be set at some limit below the VLP-16′s empirical maximum range of 120 m. In the authors’ experience, this has been found to produce vastly more accurate results. Bear in mind that this finding has been confined to primarily research missions where maximizing area of coverage was not a concern. This concern is a product of range-dependent error and generalizes to all lidar sensors. If time and resources permit, narrowing the field of view should be considered.

The factor that ensures the most evenly dispersed scan pattern when using the fan-style lidar sensor is yaw, i.e., crabbing the scanner with respect to the direction of flight (see Section 3.3). Intentionally misaligning the lidar sensor’s rotational axis and the airframe’s roll axis when mounting the payload will yield this result at minimal cost to the width of the lidar swath. This is not an option to the end user with a turnkey UAS lidar system in hand; to the authors’ knowledge, most UAS lidar payloads are designed such that the rotation axis of the scanner and the roll axis of the airframe are parallel. One possible solution is to exploit wind; planning a mission’s flight lines to be perpendicular (or at least not nearly parallel) to the direction of wind will force the UAS autopilot to crab the airframe in order to stay on course.

## 4. Conclusions

The fan-style lidar sensor (Figure 1) has become widely adopted for UAS lidar mapping. The fan-style geometry of lasers in this class of sensors creates a unique pattern of returns which warrants careful study for the purpose of planning successful UAS lidar data collection missions. For this study, an often-used fan-style lidar sensor, the Velodyne VLP-16, was selected. Through the development and use of a novel VLP-16 simulation software, the study examines the sensor’s scan pattern of the VLP-16 both qualitatively—i.e., manually examining simulated point clouds—and quantitatively, via spatial statistics. This study also presents novel characterization equations for the VLP-16′s point density, optimal separation of flight lines, and possible gaps in coverage, all of which can be generalized to other fan-style lidar sensors. Flight planning recommendations for minimizing coverage gaps are offered based on examination of the results. The factor found to have the greatest effect in minimizing the deleterious effects of the coverage gaps is the addition of yaw, i.e., crabbing the sensor with respect to the direction of flight.

## Figures and Tables

**Figure 1 sensors-20-07351-f001:**
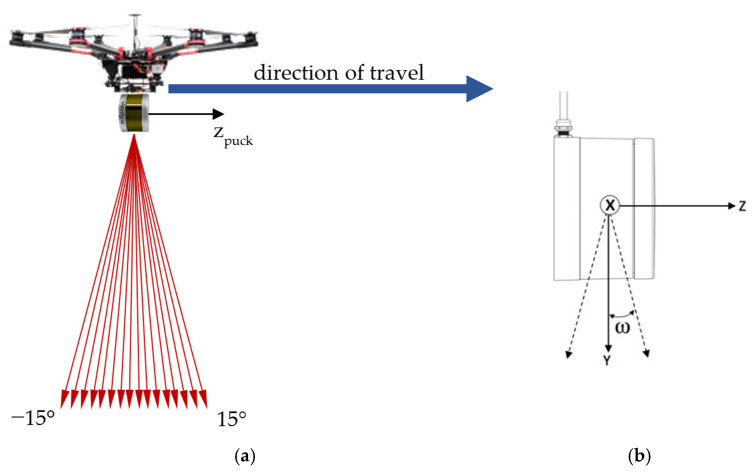
(**a**) Typical orientation of the VLP-16 scanner aboard an unoccupied aircraft system (UAS). (**b**) Scanner-oriented coordinate system of the VLP-16 (figure courtesy of Velodyne Lidar).

**Figure 2 sensors-20-07351-f002:**
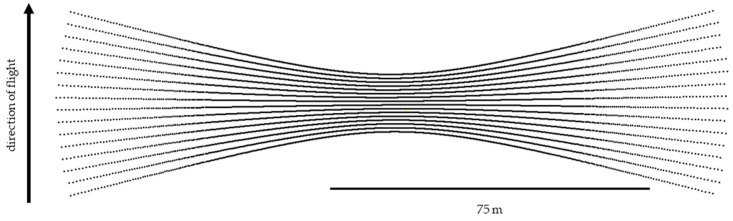
The scan pattern of a single pass of the VLP-16′s sixteen lasers over a target plane at a flying height of 30 m. Each laser traces a path of a hyperbola across the plane, and the forward motion of the scanner adds a slight affine distortion to each hyperbola.

**Figure 3 sensors-20-07351-f003:**
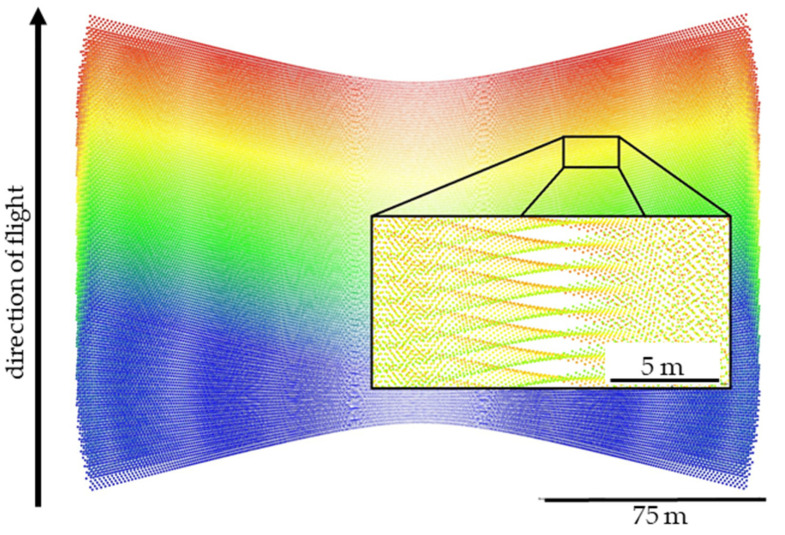
Gaps in the VLP-16 scan pattern. A simulated point cloud of the VLP-16 flying 30 m above a target plane at a speed of 10 m/s is shown. The detail shows an example of the gaps in the point coverage that result from the unique scan pattern. The above diamond-shaped gaps are approximately 1 m × 3 m each. The points are colored by time of pulse (blue > green > yellow > red).

**Figure 4 sensors-20-07351-f004:**
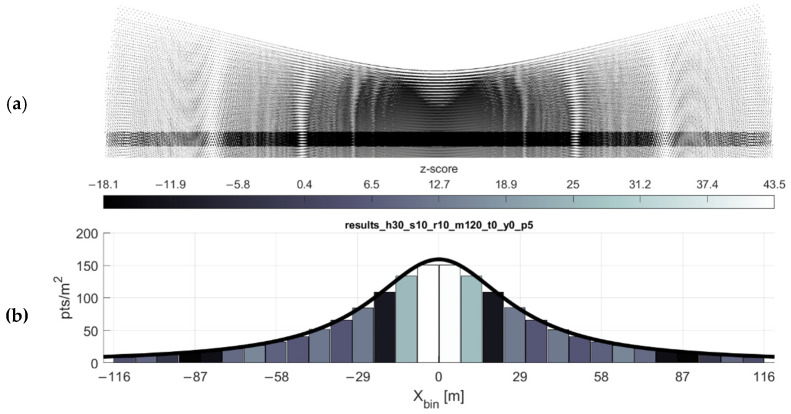
Statistical analysis of the VLP-16 scan pattern. (**a**) Because the VLP-16 scan pattern is repeating, a representative across-track profile of the scan pattern can be automatically extracted from the software to use for spatial statistical analysis. The black stripe above depicts a 5-m-wide profile in the direction of flight. (**b**) The histogram depicts the location and width of the bins. Each bin is colored by its clustering z-score (low:black::high:white). Note: For this figure, wider bins were used to better illustrate the binning concept; for the study, 1-m wide bins were used.

**Figure 5 sensors-20-07351-f005:**
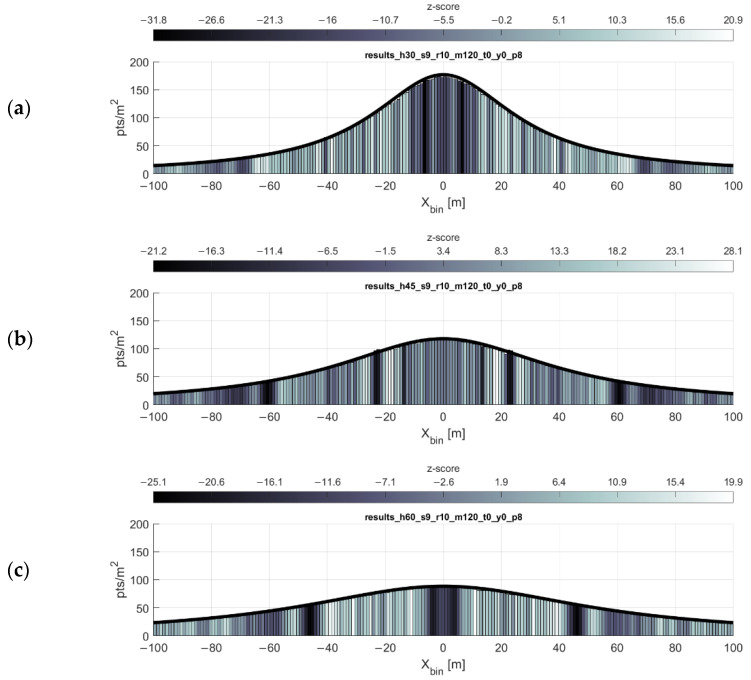
Point density function relationship to flying height. Plots show density function (black line) and simulation results (histogram) for flying heights of (**a**) 30 m, (**b**) 45 m, and (**c**) 60 m. Forward velocity held at 9 m/s.

**Figure 6 sensors-20-07351-f006:**
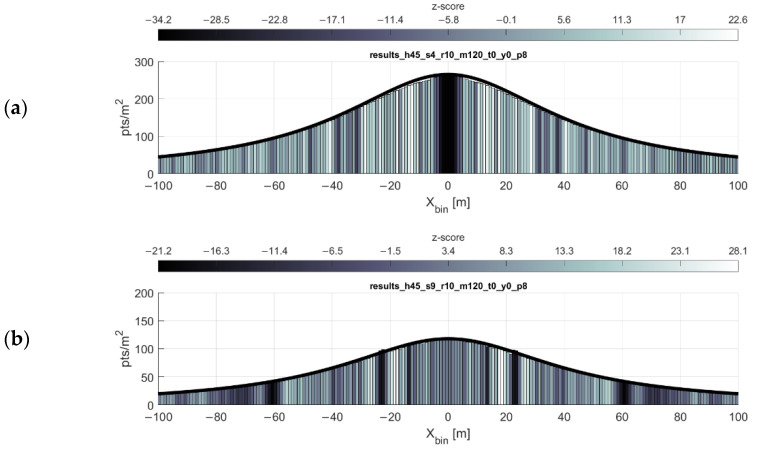
Point density function relationship to forward velocity. Point density function (black line) and simulation results (histogram) for forward speeds of (**a**) 4 m/s, (**b**) 9 m/s, and (**c**) 15 m/s. Flying height is held at 45 m. Note the y-axis difference in the topmost histogram.

**Figure 7 sensors-20-07351-f007:**
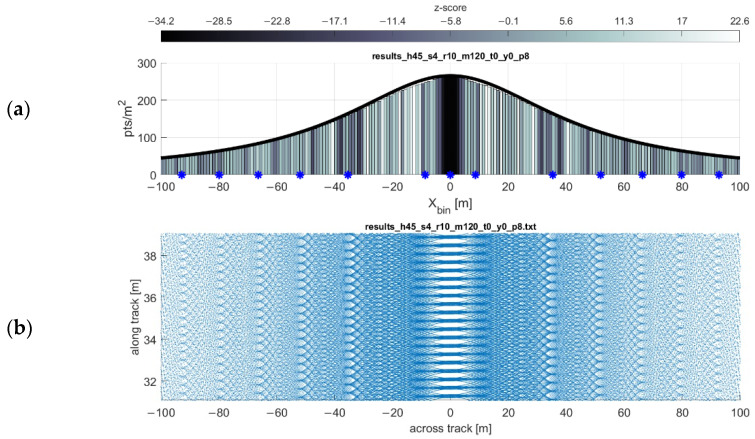
Plot of gap equation results. Flying height 45 m, forward velocity 4 m/s. (**a**) Gap locations reported by the equation are marked with blue asterisks. (**b**) Return pattern of the same flight, exaggerated in the y-direction to show detail.

**Figure 8 sensors-20-07351-f008:**
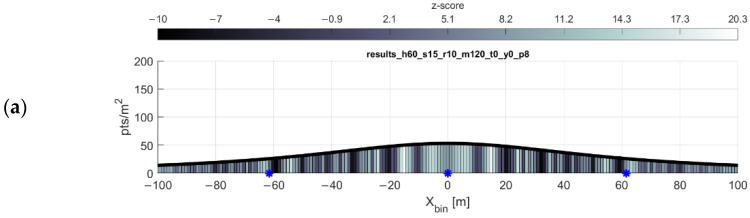
Plot of gap equation results with missing gap results. Flying height 60 m, forward velocity 15 m/s. (**a**) Gap locations reported by the equation are marked with blue asterisks. (**b**) Return pattern of the same flight, exaggerated in the y-direction to show detail.

**Figure 9 sensors-20-07351-f009:**
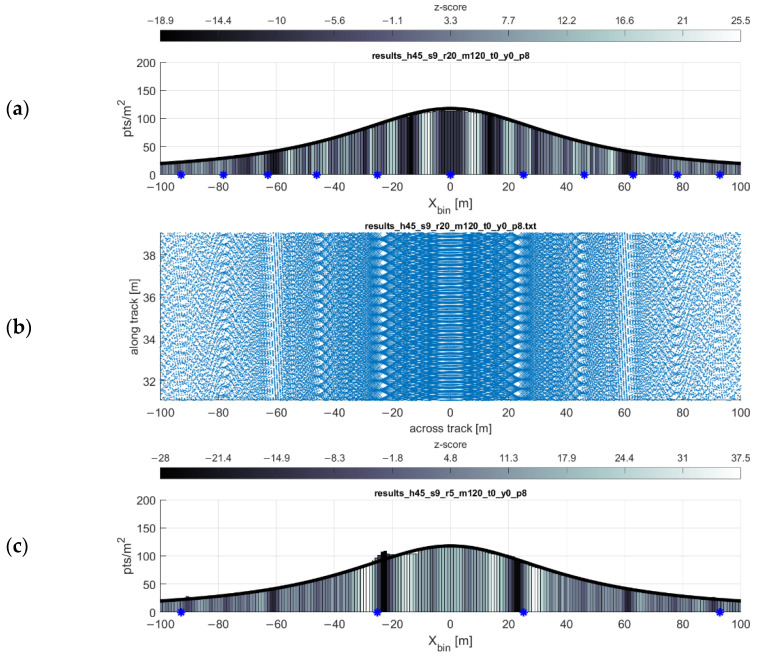
Effect of rotation rate on scan pattern and gap bands. Though the number of predicted gaps decreases as the rotation rate of the scanner head decreases, the severity of the gaps becomes notably greater. (**a**,**b**) Rotation rate 20 Hz. (**c**,**d**) Rotation rate 5 Hz. For both simulations, flying height 45 m, forward velocity 9 m/s. The histograms (**a**,**c**) and scan pattern plots (**b**,**d**) are outputs from the simulation software. From the characterization equations, the locations of the predicted gaps are shown on (**a**,**c**) as blue asterisks on the histograms, and the point density function is shown as a black line.

**Figure 10 sensors-20-07351-f010:**
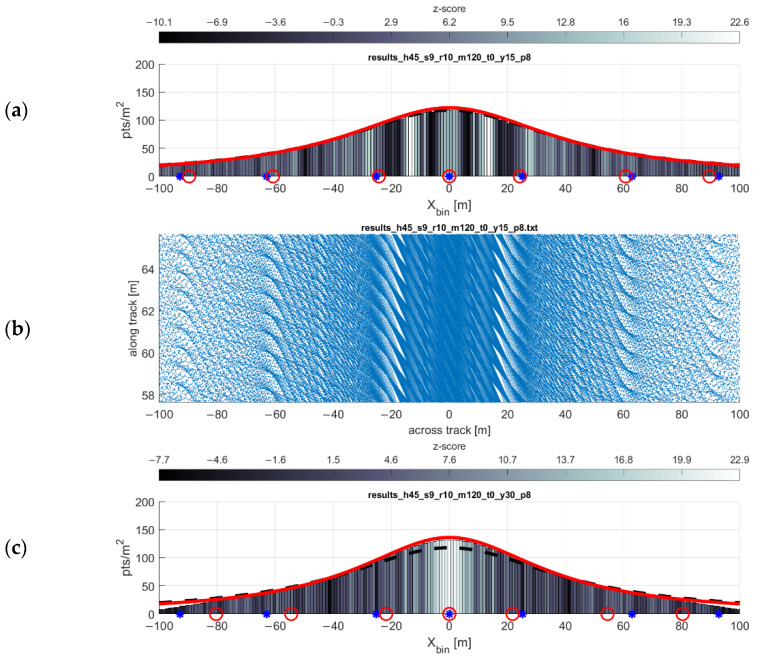
Effects of (**a**,**b**) 15° of yaw and (**c**,**d**) 30° of yaw on point density and gap locations. For (**a**,**c**), the results of the PDF and gap equation with no yaw are plotted in black and blue, respectively. The results from the PDF and gap equations plotted in red.

**Figure 11 sensors-20-07351-f011:**
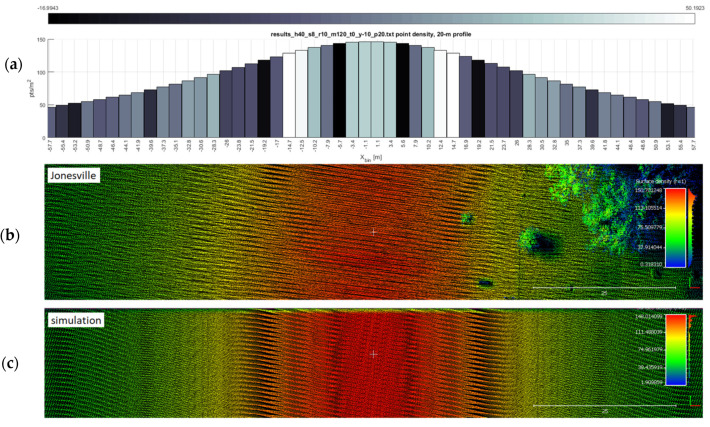
Ground truth of a flight line with nonzero yaw. This figure depicts the results of a test flight (**b**) that was flown with approximately −10° of yaw, versus the simulated point cloud (bottom) of that same mission. Both the histogram (**a**) and simulation (**c**) verify the point density, and the dark bars in the histogram correctly indicate the presence of point clustering (gaps).

**Figure 12 sensors-20-07351-f012:**
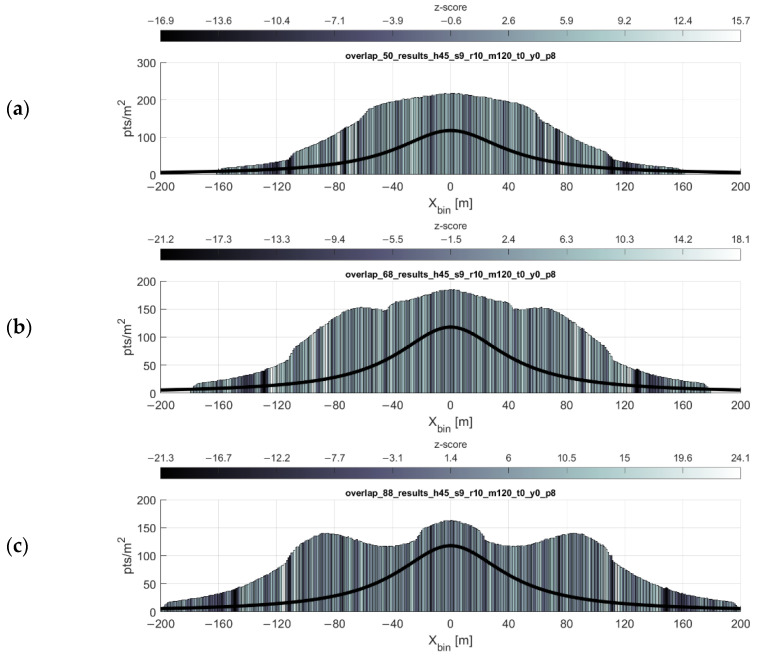
Plot of overlapping flight lines. Flying height 45 m, 9 m/s forward velocity, with parallel flight lines spaced at (**a**) 50 m, (**b**) 68 m, and (**c**) 88 m. These flight line spacings are the results of the flight line separation equation for values of minimum point density $p_{d}=$ 180, 150, and 120 points/m^2^ respectively. Point density function of a single strip shown (black line). Note the y-axis difference in the topmost histogram.

**Figure 13 sensors-20-07351-f013:**
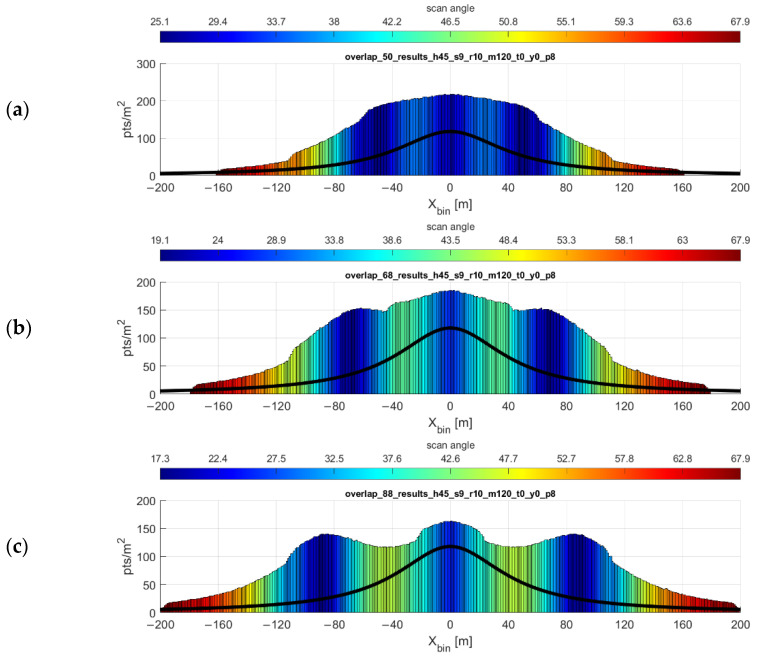
Average scan angle per bin. Flying height 45 m, 9 m/s forward velocity, with parallel flight lines spaced at (**a**) 50 m, (**b**) 68 m, and (**c**) 88 m. (The reason for these flight line spacings is explained in the caption for Figure 12). Note the y-axis difference in the topmost histogram and minor differences in the legend.

**Figure 14 sensors-20-07351-f014:**
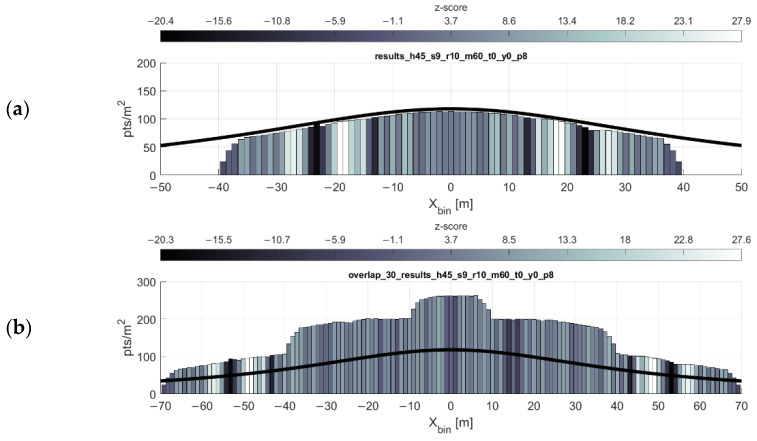
Mission planning for artificially limited maximum range. The most noted effect of limiting the maximum range is the drastically decreased profile width. (**a**) Single flight lines, flying height 45 m, forward speed 9 m/s, max range 60 m. (**b**) Overlapping flight lines, 30 m spacing, colored by nearest neighbor z-score. (**c**) Overlapping flight lines colored by average scan angle; note the change in the scale of the y-axis.

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
