# Peer review of "Scan Pattern Characterization of Velodyne VLP-16 Lidar Sensor for UAS Laser Scanning"

_sensors, 2020, doi:10.3390/s20247351_

Round 1
Reviewer 1 Report
The paper reads very well in the current form. However, it would be more interesting, if systematic errors of the Velodyne sensor are included in the simulation.
Author Response
Point 1: The paper reads very well in the current form. However, it would be more interesting, if systematic errors of the Velodyne sensor are included in the simulation.
Response 1: I agree that a thorough sensor modeling of the VLP-16 would include the option to simulate both an uncalibrated and calibrated sensor. I added a brief justification in section 2.2 for our choice of a simplified sensor model for this particular study.
Reviewer 2 Report
This manuscript discussed the scan patterns of Velodyne VLP-16 Lidar Sensor for UAS Laser Scanning. Authors developed sensor modeling software for scanning simulation and analyzed reasons for some areas of poor laser pulse coverage when using a fan-style sensor. Overall, the novelty and quality of this manuscript make it possible for publication in this journal.
I suggest the authors can add more details about the simulation software because it is a key method part for this manuscript. At least, the principle of the simulation should be clearly stated.
Below are some detailed comments:
1 In figure 2 and figure 3, can you add a scale bar and also indicate the flight direction?
2 Figure 4 and rest of figures, use figure (a), (b)… for subfigures instead of TOP, BOTTOM….
3 Figurer 5, use the same horizontal scales for all subfigures.
Author Response
Point 1: I suggest the authors can add more details about the simulation software because it is a key method part for this manuscript. At least, the principle of the simulation should be clearly stated.
Response 1: I have updated section 2.2 with further explanation of the method used to solve for the simulated laser returns and a justification for the chosen sensor model. I hope these address your concerns about the principles of the simulation.
Point 2: Below are some detailed comments:
1 In figure 2 and figure 3, can you add a scale bar and also indicate the flight direction?
2 Figure 4 and rest of figures, use figure (a), (b)… for subfigures instead of TOP, BOTTOM….
3 Figurer 5, use the same horizontal scales for all subfigures.
Response 2: Thank you, I have made these changes.
Reviewer 3 Report
28/11/2020
Dear authors,
In the manuscript Scan Pattern Characterization of Velodyne VLP-16 Lidar Sensor for UAS Laser Scanning you introduce sensor modeling software for scanning simulation and analytical equations developed in-16 house to characterize the non-uniform return density (i.e. scan pattern) of the fan-style sensor, with 17 special focus given to a popular fan-style sensor, the Velodyne VLP-16 laser scanner.
The theme is actual and interesting. The manuscript is well written, Research and analysis are done appropriately. However, the state of the art is not sufficiently covered. The condition of the area should be better covered. There are too few references mentioned in the paper, and you need to give a better insight into the area.
Be more specific when writing a Conclusion. Support your conclusions with metrics (results), there is no place for speculation (like term ‘most likely’).
Such manuscripts should be written in the third person. Please apply it to the entire text.
I suggest going through the text one more time and responding to my comments in the text, for the purpose of improving the text. It is necessary to make changes to the text, in order to improve the quality of the text.
Best regards

Author Response
Point 1: The theme is actual and interesting. The manuscript is well written, Research and analysis are done appropriately. However, the state of the art is not sufficiently covered. The condition of the area should be better covered. There are too few references mentioned in the paper, and you need to give a better insight into the area.
Response 1: I have included a number of new sources to give a better insight into the state of the art of analyzing the characteristics of multi-beam lidar sensors. There are not many sources to choose from (at least two of my sources echo this!), but your insistence to better cover the state of the art led me to a unique study that I had not found in my first literature review. This study supports some of our study's findings, and is the only other study I can find that analyzes the scan pattern of the multi-beam lidar sensor. I am very glad you suggested another look at the literature. (Sorry for the long response, I was excited to find this new study.)
Point 2: Be more specific when writing a Conclusion. Support your conclusions with metrics (results), there is no place for speculation (like term ‘most likely’).
Response 2: This is great advice. I have moved most of the old conclusion to the Results and Discussion section and I have written a new conclusion which more concisely summarizes the study.
Point 3: Such manuscripts should be written in the third person. Please apply it to the entire text.
Response 3: I have changed first-person language to third-person.
Point 4: I suggest going through the text one more time and responding to my comments in the text, for the purpose of improving the text. It is necessary to make changes to the text, in order to improve the quality of the text.
Response 4: Your comments were very helpful, and I believe the manuscript is now more clear and informative.